# Scalable Super-Resolution Neural Operator

## ABSTRACT

Recent advances in continuous super-resolution (SR) has made a substantial progress towards universal SR models, which are characterized by using a single deep neural network (DNN) to fulfill arbitrary scale SR tasks. When deployed on resource stringent platforms, however, a trained DNN model usually requires experience-demanding and laborious manual efforts to compress the models following a predetermined compute budget. This paper proposes an inference-time adaptive network width optimization method for arbitrary scale SR modules, dubbed as Scalable Super-Resolution Neural Operator (SSRNO), which is capable of efficien performance-preserving deployment on various mobile or edge devices with only a user input parameter indicating the desired compression rate. SSRNO realizes the continuous parameterization of SRNO[42] by virtue of two novel contributions. First, we propose the Integral Neural Network (INN) formulation for the Galerkin type attention, which is an indispensable component for spatial discretization invariant SR neural networks. Second, we further propose an adaptive layer-wise compression rate estimation mechanism, which allows for the flexible adaptation to variant capacity through the neural network layers. Extensive experiments validate the outperforming overall performances over existing continuous SR models in terms of reconstruction accuracy, model scalability as well as throughput. For instance, compared with the baseline SRNO, a typical configuration of SSRNO can achieve a model size compression up to 62% and an over 2× speedup in situations where resources are limited, while it can also expand itself to keep the PSNR degradation within 0.1 dBs when the limitations are alleviated. The code will be made public soon.

## CCS CONCEPTS

• **Computing methodologies** → **Image representations**; **Reconstruction**; **Image processing**; **Antialiasing**.

## KEYWORDS

Super-resolution, Arbitrary Scale, Neural Network Compression, Galerkin Attention

## 1 INTRODUCTION

Single-image super-resolution (SISR) is a computer vision task that reconstructs a high-resolution (HR) image from a low-resolution (LR) image, which has high practical utility in many fields like security cameras[1, 35], medical diagnosis[11], object detection[37] *etc.*

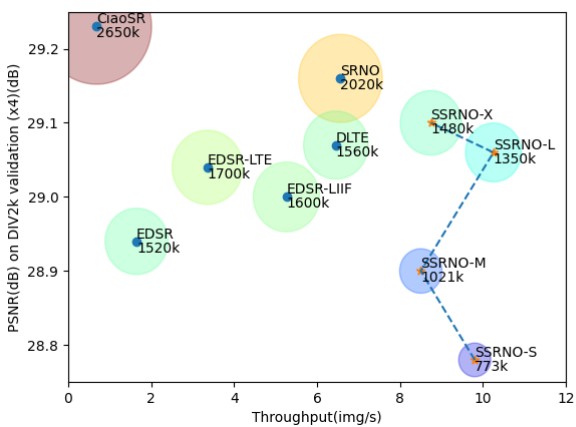

**Figure 1: Comparison of continuous SR models on DIV2K validation (×4) for PSNR, model size (MB) and speed. Throughput is tested on Urban100(×4, image size 256×161).**

Early deep-learning based image super-resolution (SR) methoda [9, 15, 22, 27, 33] use convolutional neural networks (CNNs) to approximate nonlinear mappings between LR-HR image pairs. As manifested by the successful applications ranging from natural language processing to computer vision, the attention mechanism of transformers[41] has also been explored in SISR tasks, and further augmented with Implicit Neural Representation [10, 38], achieving appealing arbitrary scale. Reconstruction results with a single module[7, 12, 17, 36, 42]. Particularly, Super-Resolution Neural Operator (SRNO)[42] makes use of the galerkin-type attention[8] to learn the operator mapping between the LR and HR image function spaces, allowing for continuous and zero-shot super-resolution. However, most of these methods have complex DNNs structures with large amount of parameters, which leads to unfordable consumption of computational resources when deployed on edge devices.

Several recent works[25, 30] construct efficient transformer or attention structures to reduce the parameters and FLOPs. As analyzed in[25, 30] and also verified in our experiments, muti-head self-attention (MHSA) is usually less efficient than feed-forward network (FFN) layers. MHSA heads are not of equal importance and there exists high similarities among attention heads, which means some heads learn similar projections of the input full features and thus gives rise to computational redundancy. Moreover, due to the qudratic computational compelxity, MHSA runs much slower than FFN. Therefore, many works focus on reducing unimportant attention heads[19] or change the attention structures[43] to make MHSA more efficient. Nevertheless, these methods usually require manual adjustment of the target compression rate and careful pruning procedures, which can involve repeated fine-tuning

jobs demanding experiences and skills. What makes these methods more undesirable is that they can only produce a single compressed model and can not provide the flexibility to choose the appropriate model size based on the target platform resources to achieve the best performance.

Integral Neural Networks (INN)[39] is a newly emerged novel model compression method for convolutional layers and fully-connected layers, which are represented as integrals of continuous weight functions. Practically, the number of sampled points from those continues functions determines the number of parameters in resulting models. In order to maintain the performance over different model sizes, INN trains various size of parameter partition of the model weights, and requires a user input compression rate to consistently sample each layer of the continuously parameterized underlying DNN.

Although INN performs well on convolution layers and fully-connected layers, the continuous parameterization for the attention layer in transformer blocks[41] has not been addressed by INN[39]. The difficulty mainly arises from the soft-max operation (see analysis in Sec. 2.3), which hinders its application to the prevalent Transformer-based DNN compressions. Moreover, the choice of using the same compression rate for the whole model is suboptimal. Without carefully considering the range of compression rates, the performance of INN can significantly degrade because different layers in DNNs have different redundancies [6, 16, 32] and the total compression budget should be able to adaptively distributed among different layers.

To address the above issues, we propose Scalable Super-Resolution Neural Operator (SSRNO), which achieves the continuous parameterization of SRNO[42] and therefore is capable of adaptive DNN scalability at inference time while keeping the model performance. First, we propose the INN formulation for the Galerkin type attention, which is an indispensable component for the spatial discretization invariant neural networks. We further propose an adaptive layer-wise compression rate estimation mechanism, which allows for the flexible adaptation to variant capacity among neural network layers. Our method mitigates the impact of human expertise on model compression and provides stable optimization results. Our experiments show that SSRNO achieves up to 86% compression rate in the attention part, and a minimum model size of 38% leading to around 2× through-put on Nvidia RTX 4090 GPU, Intel Xeon CPU and the ONNX format, while the performance loss in PSNR can be controlled below 0.1 dB. Fig.1 presents a comparison of arbitrary scale SR models in terms of accuracy, model size and inference speed.

In summary, our main contributions are as follows:

- We propose the INN representation for the Galerkin-type attention mechanism, which completes the toolbox for continuous parameterization of attention-based DNNs.
- We propose an addaptive layer-wise compression rate estimation method, which can automatically select the optimal layer compression rate.
- We train and verify the Scalable Super-Resolution Neural Operator (SSRNO) for arbitrary scale SR problems, which for the first time fulfills compute-agnostic arbitrary scale

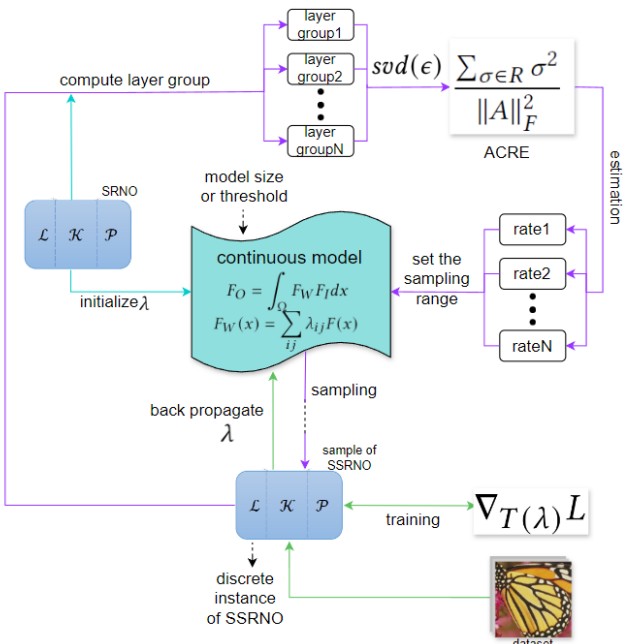

Figure 2: Workflows of SSRNO. The solid lines represent the model training process, while the dashed line denote the inference process. A discrete instance of SSRNO can be sampled by setting a single hyper-parameter of model size or threshold epsilon. See Sec. 3.3 for more details.

SR model training and flexible model size adjustment at inference time.

## 2 RELATED WORK

### 2.1 Super-Resolution Neural Operator

Neural Operator (NO) is a novel neural network architecture proposed for discretization invariant solutions of PDEs via infinite dimensional operator learning[26]. The goal of NO is to learn a mapping $\mathcal{G} : \mathcal{A} \rightarrow \mathcal{U}$ between two infinite dimensional spaces $\mathcal{A}$ and $\mathcal{U}$ by using a finite collection of observations of input-output pairs. In practice, NO employs a three-phrase architecture to approximate $\mathcal{G}$ by a neural network $G_\theta$ with $\theta$ the parameters:

$$z_0(x) = \mathcal{L}(x, a(x))$$
$$z_{t+1}(x) = \sigma(W_t z_t(x) + (\mathcal{K}_t(z_t; \Phi))(x)) \qquad (1)$$
$$u(x) = \mathcal{P}(z_T(x)),$$

where $\mathcal{L} : \mathbb{R}^{d_a+d} \rightarrow \mathbb{R}^{d_z}$ is the lifting function mapping input $a$ to the first hidden representation $z_0$ and $\mathcal{P} : \mathbb{R}^{d_z} \rightarrow \mathbb{R}^{d_u}$ is the projection function projecting the final output of the kernel integration back to the output function space $\mathcal{U}$. $W : \mathbb{R}^{d_z} \rightarrow \mathbb{R}^{d_z}$ is a linear transformation and $\sigma : \mathbb{R}^{d_z} \rightarrow \mathbb{R}^{d_z}$ is the nonlinear activation function. For the integration kernel $\mathcal{K}_t : \{z_t : D_t \rightarrow \mathbb{R}^{d_z}\} \mapsto \{z_{t+1} : D_{t+1} \rightarrow \mathbb{R}^{d_z}\}$, one of the most adopted forms is :

$$(\mathcal{K}_t(z_t; \Phi))(x) = \int_D K_t(x, y; \Phi) z_t(y) \mathrm{d}y, \quad \forall x \in D \qquad (2)$$

where the kernel function $K_t : \mathbb{R}^{d+d} \rightarrow \mathbb{R}^{d_z \times d_z}$ is parameterized by $\Phi$.

SRNO utilizes the NO framework to solve the SR problem. The goal of SRNO is to learn an operator between LR and HR image spaces, where image instances are regarded as evaluations of continuous functions on specified locations. By learning the mapping relationship between different magnification ratios, SRNO can transform LR images into images of arbitrary resolutions. In implementation, SRNO uses the same architecture with NO as in Eq.1. For the dual consideration of model size and runtime efficiency, SRNO utilizes Galerkin-type attention as the kernel integration which will discussed next.

## 2.2 Galerkin Attention

Galerkin attention is a novel self-attention mechanism and it is a linear variant of fourier-type attention without the softmax normalization. The galerkin-type attention is formed as follows:

$$y = y + g(y + Attn(y))$$
$$Attn(y) = Q(\tilde{K}^T \tilde{V})/n \qquad (3)$$
$$\tilde{\diamond} = Ln(xW_\diamond)$$

where $W_\diamond \in \mathbb{R}^{d \times d}, \diamond \in \{Q, K, V\}$ are the trainable projection matrices, $n$ is sequence length, $g(\cdot)$ refers to a standard 2-layer FFN and $Ln(\cdot)$ refers to the layer normalization[4]. The vanilla attention has a complexity of $O(n^2 d)$ where $n$ is the token number. The matrix product $QK^T$ as well as the subsequent soft-max intermediate result are both needed to be stored and take $O(n^2)$ memroy[14]. In contrast, galerkin-type attention is softmax-free and compute $K^T V$ first. It only has $O(nd^2)$ time complexity and needs $O(d^2)$ storage. As usually $n \gg d$ in super-resolution problems, galerkin attention is more time and memory-efficient. Furthermore, the elimination of the soft-max operation in Galerkin-type Attention reduces the GPU memory access latency, which in turn contributes to improvement on the running speed[13, 14].

## 2.3 Integral Neural Network

INN proposes the continuous parameterization method for the convolution layers as well as fully-connected layers. For the convolution layer, the integral representation is formulated as follows:

$$F_O(x^{out}, x^{s'}) = \int_\Omega F_W(\lambda, x^{out}, x^{in}, x^s) F_I(x^{in}, x^s + x^{s'}) dx^{in} dx^s \qquad (4)$$

where $F_I(\cdot, \cdot), F_O(\cdot, \cdot)$ are integrable functions representing the input and output images, $x^s$ represents the spatial dimension corresponding to the convolution operation, while $x^{in}$ represents channel dimension. Just as how convolutional neural networks work in practice, INN uses Eq.4 to perform simultaneous integration over both spatial and channel dimensions, where the trainable parameters $\lambda$ get optimized during training. For the fully-connected layer, the integral representation is formulated as follows:

$$F_O(x^{out}) = \int_\Omega F_W(\lambda, x^{out}, x^{in}) F_I(x^{in}) dx^{in} \qquad (5)$$

where the integral is performed along the channel dimension. The parametric kernel function $F_W$ in the integral representation is formulated as:

$$F_W(\lambda, x^{out}, x^{in}) = \sum_{i,j} \lambda_{ij} u(x^{out} m^{out} - i) u(x^{in} m^{in} - j) \qquad (6)$$

where $u(\cdot)$ is the cubic convolutional kernel. Via Eq.6, INN constructs a continuous neural network, meaning that the shape of neural network can be modified by appointing a different discretization of the integral kernel. For the typical neural operator, the integral form is as:

$$u(x) = \sigma\left(v(x) + \int_D \mathcal{K}_v(x, y, v(x), v(y)) v(y) dy\right)$$

$$\mathcal{K}_v(v(x), v(y)) = \int_D \frac{exp(\int_\Omega Q_t v(x,t) K_t v(y,t) dt)}{\int_D exp(\int_\Omega Q_t v(s,t) K_t v(y,t) dt) ds} V v(y) dy \qquad (7)$$

where $v(\cdot)$ refers to the continuous input, Q/K/V are both the trainable parameters. Due to the use of soft-max in integral kernel, the denominator of the kernel involves the integration of both channel and spatial dimensions, which are both suffering from the sampling error of the INN method, leading to the accumulation of errors. Furthermore, the exponential form also amplifies the impact of sampling errors on the original data distribution. Therefore, the nested integral form included in Eq.7 makes the error in the integral calculation uncontrollable and it is challenging to convert the this into continuous format with a good preference.

## 3 METHOD

We first present the integral representation for the Galerkin attention, then propose an adaptive compression rate estimation algorithm for layer width determination, paving the way for the continuous parameterization of SRNO.

## 3.1 Scalable Galerkin-type Attention

Galerkin-type attention utilizes layer normalization operation instead of soft-max, which can be formulated as:

$$LN(x)_{ij} = \frac{x_{ij} - \mu}{\sigma} \times \gamma + \beta \qquad (8)$$

where $\mu$ is the mean value ,$\sigma$ is the standard deviation and $\gamma, \beta$ are trainable parameters. Unlike the softmax operation, the sampling error of $x_{ij}$ only makes particular influence on its own output but contributes equally to other output items of the LN operation, indicating the continuity and stability. It is straightforward to convert the LN operation into the integral formulation.

In the word of MHSA, the Q/K/V transformations as integral transform as follows:

$$F_\diamond(x_o^d, x_o^p) = \int_\Omega F_{W_\diamond}(\lambda, x_o^d, x_o^p, x_i^p) F_x(x_i^p, x_i^d) dx_i^p dx_i^d \qquad (9)$$

where $\diamond \in \{Q, K, V\}$, $\lambda$ is the trainable parameters which can be initialized by the $W_Q/W_K/W_V$ parameter matrices, $x^p \in \mathbb{R}^{n_x \times n_y}$ refers to the spatial coordinate, $x^d \in \mathbb{R}^d$ refers to the feature coordinate and $F_x(\cdot)$ represents the continuous input signal. For the $x_o^d$-th head of MHSA, the integral in Eq.9 map the continuous input signal $F_x(\cdot)$ to Q/K/V hidden spaces $F_{Q(KV)}(x_o^d, \cdot)$. The scalable

Galerkin attention part can be written as:

$$\mathbf{sga}(x^d, x^p) = \int_{y^d \in \Omega} F_Q(x^p, y^d) \int_{y^p \in D} \tilde{F}_K(y^p, y^d)\tilde{F}_V(y^p, x^d)dy^p dy^d \tag{10}$$

where the $\tilde{F}_\Diamond$ refers to the layer normalization on $F_\Diamond$ with $\Diamond \in \{K, V\}$, which can be computed with Eq.8.

## 3.2 Adaptive Compression Rate Estimation

A common way for setting the channel dimension hyper-parameters is to follow the configurations in standard transformer models[24, 31, 40, 42], which often set the same value to all layers. However, as different model compression tasks are faced with different computational resource budget, the model compression settings should be able to change accordingly. Furthermore, the parameter requirements of network capacities vary across different layers[16, 32]. To address this issue, we propose an adaptive layer-wise compression rate estimation mechanism summarized in Algorithm 1.

---

**Algorithm 1:** Adaptive Compression Rate Estimation (ACRE)

| | |
|---|---|
| **input** | : network $G$ |
| **hyperparameter** | : threshold $\epsilon$ |
| **required functions** | : function $SVD(\cdot)$ return the singular values, descending sort algorithm $sort(\cdot)$ |

1 **for** *layer group LG in G* **do**
2   **for** *layer l in LG* **do**
3     **First step**: get the dispersed matrices
4     **for** *parameter matrix X in l* **do**
5       //X is a four dimensional tensor shape of (n,m,k,k)
6       //m for the input channel, n for the output channel and k for the kernel size
7       **if** *l is attention-type layer* **then**
8         $mats \leftarrow mats + X[:,:,0,0]$
9       **else**
10         **for** $i \leftarrow 0$ *to* $k$ **do**
11           **for** $j \leftarrow 0$ *to* $k$ **do**
12             $mats \leftarrow mats + X[:,:,i,j]$
13     **Second step**: estimate the ranks
14     **for** *matrix Y in mats* **do**
15       $v \leftarrow sort(SVD(Y))$
16       **foreach** *singular value e in v* **do** $m \leftarrow m + e^2$;
17       **for** *i in v* **do**
18         $s \leftarrow i^2 + s$
19         **if** $s/m > \epsilon$ **then** $Y.r \leftarrow i$,break;
20     **Third step**: estimate the compression rate
21     **for** *matrix Y in mats* **do**
22       $LG.rate = min(L.rate, Y.r)$

---

Noticed that small singular value represent the compression of space by matrix transformations, the singular value can be used to

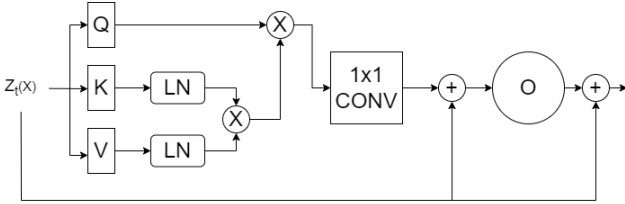

**Figure 3: The visualization of modified Galerkin-type attention architecture.**

represent the significance or importance of a space, referring to the potential redundancy of the space. So we utilize a method based on the Frobenius norm to estimate the redundancy of the matrix based on its singular values. For any parameter matrix $A \in \mathbb{R}^{m \times n}$, the Frobenius norm of $A$ is defined as

$$\|A\|_F = \sqrt{\sum_{i=1}^{m}\sum_{j=1}^{n}|A_{ij}|^2} = \sqrt{\sum_{i=1}^{min(m,n)}\sigma_i^2} \tag{11}$$

where $\sigma$ refers to the singular value of $A$. We define the estimation quality M as follow:

$$M = \frac{\sum_{\sigma \in R}\sigma^2}{\|A\|_F^2} \tag{12}$$
$$R \subset \{\sigma | \sigma \text{ is the singular value of } A\}$$

With the definition of the estimation quality, we can choose a threshold $\epsilon$ manually and define the estimated rank. The estimated rank r of matrix A can be represented as:

$$r_\epsilon = |R|, \min_r M > \epsilon \tag{13}$$

So, the redundancy of A can be defined as $\frac{r}{min(m,n)}$.

For the three-order tensor $\mathbf{y} \in \mathbb{R}^{m \times n \times t}$, which corresponds to the case of multichannel convolution we adopt the same operation as the two-dimensional case, which corresponds to FFN layers.. We expand the tensor $\mathbf{y}$ along dimension $t$ and treat the result matrices as two-dimensional matrix to compute their redundancy respectively. We choose the minimum redundancy of these $t$ matrices as the redundancy of tensor $\mathbf{y}$. We adopt this approach because we consider that the same relative positions across different convolutional kernels should have the same level of redundancy, while different relative positions may exhibit variations in redundancy.

## 3.3 Scalable Super-Resolution Neural Operator

Fig.2 depicts the workflows of SSRNO. There are two main cycles in the training process. The purple lines represent the cycle that updates the compression ranges during training, while the green lines denote the process of optimizing the model parameters $\lambda$. The continuous model is initialized with the weights in SRNO. We generate some partitions by calculating the parameter functions using Eq.6 with the number of sampling point in the corresponding compression range. The computation of these sampled models is treated as integral operation as Eq.4, Eq.5 and Eq.10. The training parameters $\lambda$ get optimized through the back propagation the supervision loss gradient. After 25 epochs of sampling and back propagating which are represented as green lines, we sample the

**Table 1: Comparison on DIV2K validation set (PSNR).**

| Method | Params | In-distribution | | | Out-of-distribution | | | | |
|---|---|---|---|---|---|---|---|---|---|
| | | ×2 | ×3 | ×4 | ×6 | ×12 | ×18 | ×24 | ×30 |
| Bicubic | - | 31.01 | 28.22 | 26.66 | 24.82 | 22.27 | 21.00 | 20.19 | 19.59 |
| EDSR-baseline [28] | 1.52M(×4),1.37M(×2) | 34.55 | 30.90 | 28.94 | - | - | - | - | - |
| EDSR-SRNO [42] | 2.02M(1.22+0.80) | 34.85 | 31.11 | 29.16 | 26.90 | 23.84 | 22.29 | 21.24 | 20.56 |
| EDSR-baseline-MetaSR [10, 20] | 1.69M | 34.64 | 30.93 | 28.92 | 26.61 | 23.55 | 22.03 | 21.06 | 20.37 |
| EDSR-baseline-LIIF [10] | 1.60M | 34.67 | 30.96 | 29.00 | 26.75 | 23.71 | 22.17 | 21.18 | 20.48 |
| EDSR-baseline-LTE [24] | 1.71M | 34.72 | 31.02 | 29.04 | 26.81 | 23.78 | 22.23 | 21.24 | 20.53 |
| EDSR-baseline-DLTE[17] | 1.56M | 34.74 | 31.04 | 29.07 | 26.82 | 23.79 | 22.23 | 21.24 | 20.53 |
| EDSR-baseline-CiaoSR[7] | 2.65M | 34.91 | 31.15 | 29.23 | 26.95 | 23.88 | 22.32 | 21.32 | 20.59 |
| SRNO-INN-Attention | 1.50M(1.22+0.28) | 34.77 | 31.05 | 29.10 | 26.84 | 23.79 | 22.25 | 21.24 | 20.54 |
| SSRNO-S(Ours) | 0.77M(0.60+0.17) | 34.38 | 30.70 | 28.78 | 26.56 | 23.58 | 22.08 | 21.10 | 20.42 |
| SSRNO-M(Ours) | 1.02M(0.74+0.28) | 34.55 | 30.84 | 28.90 | 26.66 | 23.66 | 22.14 | 21.16 | 20.47 |
| SSRNO-L(Ours) | 1.35M(1.22+0.13) | 34.74 | 31.00 | 29.06 | 26.81 | 23.77 | 22.22 | 21.23 | 20.53 |
| SSRNO-X(Ours) | 1.48M(1.22+0.26) | 34.79 | 31.06 | 29.10 | 26.85 | 23.80 | 22.25 | 21.25 | 20.54 |

The parentheses indicate the parameter distribution over the EDSR encoder and the Galerkin attention parts. SSRNO-L/M have outstanding performances when considering both model size and runtime efficiency comprehensively. The SRNO-INN-Attention refers to modify the attention part of SRNO [42] with 0.5 compression rate instead of the proposed ACRE method. Note that SSRNO-X with the same amount of parameters performs better than simply using INN's fixed compression rate.

continuous model to the maximum model size (corresponding to no compression) to update the compression range of each layer. We firstly compute the layer groups based on the relationship of their channel length to ensure the correctness of the compressed network structure. Then, we adopt the ACRE method with a threshold of $\epsilon$ to estimate the maximum compression rate of these N groups respectively. Finally, we update the compression range for each layer based on its corresponding maximum compression rate. At the inference time, we sample the SSRNO with the setting of model size or threshold. We choose the results of the parameter function as the parameter matrices of discrete SSRNO.

With the definitions and analysis above, we propose the scalable super-resolution neural operator architecture as an example to utilize our methods following by the three-phase neural operator architecture:

$$z_0(x) = \mathcal{L}(x, a(x))$$
$$z_{t+1}(x) = \sigma(\int_\Omega F_{W_{t+1}}(\lambda, x, y)z_t(y)dy + (sga_{t+1}(x^p, x^d))) \quad (14)$$
$$u(x) = \mathcal{P}(z_T(x)),$$

For the scalable galerkin attention part, we modified the original galerkin-type attention architecture as Fig. 3. We add a linear operation after the QKV transformation to decouple the input and the output of the attention part so that it is not necessary for $W_Q, W_K, W_V$ to be square matrices. After this modification, the Attention module accepts input signals of dimension $d$ and maps them to a $d'$-dimensional latent space through the QKV transformation. Although it brings the additional linear layer parameter, this will help ACRE method to find the optional compression rates in different QKV transformation parts, resulting in reducing the overall parameter count and improving computation speed.

**Table 2: Thresholds chosen for evaluation for each model.**

| Model name | encoder part | attention part |
|---|---|---|
| SSRNO-S | 0.92 | 0.92 |
| SSRNO-M | 0.97 | 0.92 |
| SSRNO-L | 1.00 | 0.92 |
| SSRNO-X | 1.00 | 0.97 |

We choose different threshold for evaluation. We select thresholds in this way to test the model's ability to adjust its magnitude.

## 4 EXPERIMENT

### 4.1 Datasets and Implementation Details

We utilize the DIV2K dataset [2] for training, while the DIV2K validation set [2], Urban100 [21], B100 [34], Set5 [5] and Set14 [44] for evaluation. Peak signal-to-noise ratio (PSNR) is used as the evaluation metric.

The models are trained for 1000 epochs with batch size 64 using the L1 loss [29]. The Adam optimizer [23] employs an initial learning rate of $3 \times 10^{-5}$ and a warm-up strategy for 50 epochs following with cosine learning rate scheduler. We use the same training strategy as SRNO[42].

We train SSRNO with a threshold of $\epsilon = 0.92$, and specify different thresholds during evaluation. Specifically, we train SSRNO-S and SSRNO-M for both the encoder part and the attention part, while SSRNO-L and SSRNO-X compress the attention part only. We sample the final continuous model with high compression rate and low compression rate to demonstrate the scalability of the model. The evaluation thresholds are chosen as Table 2. Our model is implemented using the PyTorch framework and trained on a platform with two NVIDIA 4090 GPUs and Intel Xeon 8336C CPU. We set the competition models[10, 20, 24, 28, 42] with the same

**Table 3: comparison on benchmark datasets (PSNR (dB)).**

| Method | In-distribution | | | Out-of-distribution | | In-distribution | | | Out-of-distribution | |
|---|---|---|---|---|---|---|---|---|---|---|
| | ×2 | ×3 | ×4 | ×6 | ×8 | ×2 | ×3 | ×4 | ×6 | ×8 |
| | Set5 | | | | | Set14 | | | | |
| EDSR-SRNO[42] | 38.15 | 34.53 | 32.39 | 29.06 | 27.06 | 33.83 | 30.50 | 29.79 | 26.55 | 25.05 |
| EDSR-LTE[24] | 38.03 | 34.48 | 32.27 | 28.96 | 27.04 | 33.71 | 30.41 | 28.67 | 26.49 | 24.98 |
| EDSR-CiaoSR[7] | 38.13 | 34.49 | 32.43 | 29.12 | 27.16 | 33.91 | 30.47 | 28.79 | 26.60 | 25.09 |
| SRNO-INN-Attention | 38.09 | 34.48 | 32.34 | 28.94 | 27.04 | 33.74 | 30.43 | 28.71 | 25.88 | 24.96 |
| SSRNO-S(Ours) | 37.76 | 34.09 | 31.93 | 28.58 | 26.68 | 33.40 | 30.15 | 28.42 | 26.25 | 24.76 |
| SSRNO-M(Ours) | 37.91 | 34.26 | 32.12 | 28.72 | 26.86 | 33.55 | 30.16 | 28.53 | 26.35 | 24.86 |
| SSRNO-L(Ours) | 38.05 | 34.43 | 32.27 | 28.81 | 26.98 | 33.70 | 30.40 | 28.69 | 26.48 | 24.96 |
| SSRNO-X(Ours) | 38.11 | 34.52 | 32.33 | 28.87 | 27.03 | 33.75 | 30.45 | 28.72 | 26.51 | 24.99 |
| | B100 | | | | | Urban100 | | | | |
| EDSR-SRNO [42] | 32.27 | 29.20 | 27.67 | 25.91 | 24.88 | 32.60 | 28.56 | 26.50 | 24.08 | 22.70 |
| EDSR-LTE[24] | 32.22 | 29.15 | 27.63 | 25.87 | 24.83 | 32.29 | 28.32 | 26.25 | 23.84 | 22.52 |
| EDSR-CiaoSR[7] | 32.27 | 29.19 | 27.70 | 25.94 | 24.88 | 32.78 | 28.67 | 26.68 | 24.23 | 22.83 |
| SRNO-INN-Attention | 32.21 | 29.16 | 27.64 | 25.88 | 24.84 | 32.44 | 28.41 | 26.37 | 23.95 | 22.61 |
| SSRNO-S(Ours) | 32.04 | 28.98 | 27.46 | 25.73 | 24.68 | 31.65 | 27.79 | 25.79 | 23.52 | 22.26 |
| SSRNO-M(Ours) | 32.13 | 29.06 | 27.54 | 25.79 | 24.74 | 31.95 | 28.00 | 25.97 | 23.65 | 22.36 |
| SSRNO-L(Ours) | 32.21 | 29.13 | 27.62 | 25.87 | 24.83 | 32.33 | 28.31 | 26.30 | 23.90 | 22.57 |
| SSRNO-X(Ours) | 32.24 | 29.16 | 27.64 | 25.89 | 24.84 | 32.44 | 28.42 | 26.36 | 23.94 | 22.60 |

SSRNO performs better than SRNO-INN-Attention and close to SRNO[42]. It can be observed that as the model size increases, there is a significant improvement in the model's performance. In addition, our models maintain good generalization ability.

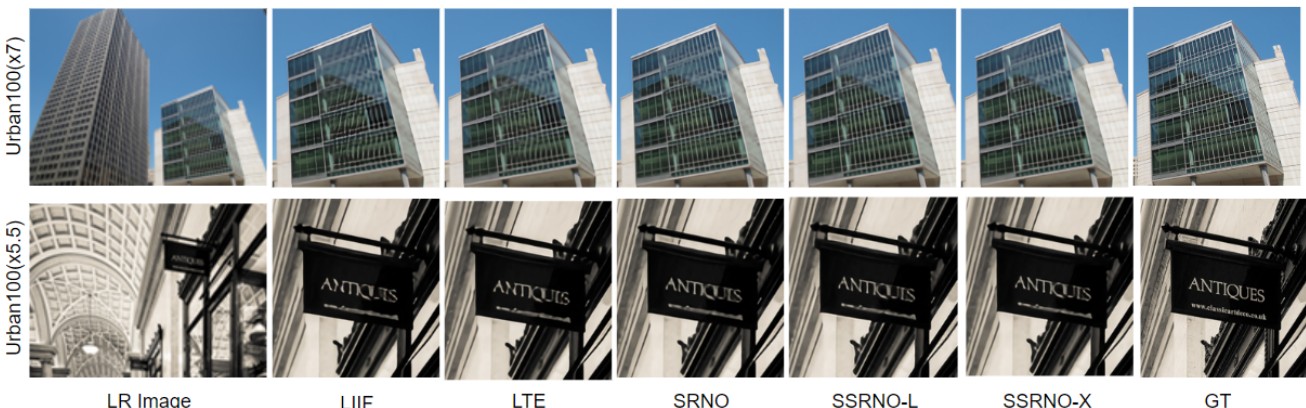

**Figure 4: Visual comparison on other zero-shot SR. We compared the performance of our method with LIIF[10], LTE[24], and SRNO[42]. Despite our model being smaller and faster, it outperforms LIIF and LTE in terms of better reconstruction of the texture integrity, thanks to the global feature capturing capability of Galerkin attention. All methods are trained with continuous random scales in ×1 − ×4 and tested on ×5.5/×7 to evaluate the generalization capability to unknown scaling factors.The EDSR-baseline is used as the encoder.**

configurations as their original paper suggested and initialize them with the provided pre-trained model.

## 4.2 Quantitative Result

Table 1 presents a quantitative comparison with other continuous super-resolution models[7, 10, 17, 20, 24, 28, 42] on the DIV2K validation dataset. It is shown that our model has the minimum amount of parameters while it performs well. The SSRNO-L achieves a total

compression rate of 0.86 on the galerkin-attention layer and has a parameter count of only 1.35 millions maintaining a performance gap within 0.1db compared with SRNO model. We outperform other SR models[10, 20, 24] with lower and flexible model size. We further compare our model with other efficient arbitrary SR models[7, 17]. Our largest model performs better than DLTE although our model size is smaller. CiaoSR performs slightly better than our models but CiaoSR is almost 1.8 times large in parameter count and consumes

Table 4: Speed comparison on 100 image in ×4 Urban100 (256×161).

| Method | #Params. | #FLOPs | Runtime(s) | | | |
|---|---|---|---|---|---|---|
| | | | ONNX | GPU | GPU_InplaceGelu | CPU |
| EDSR-SRNO [42] | 2.02M | 571.03G | 517.26 | 15.23 | 14.25 | 848.82 |
| EDSR-LTE [24] | 1.71M | 583.685G | 690.23 | 29.80 | - | 894.23 |
| EDSR-LIIF [10] | 1.57M | 962.52G | 582.32 | 19.00 | - | 798.00 |
| EDSR-CiaoSR [7] | 2.65M | 1885.98G | - | 146.16 | - | 4209.60 |
| SRNO-INN-Attention | 1.50M | 202.83G | 314.94 | 11.74 | 8.02 | 758.40 |
| SSRNO-S(Ours) | 0.77M | 136.52G | 228.09 | 10.20 | 6.93 | 715.50 |
| SSRNO-L(Ours) | 1.35M | 134.07G | 233.88 | 9.75 | 6.16 | 685.14 |
| SSRNO-X(Ours) | 1.48M | 222.07G | 289.02 | 11.43 | 8.39 | 871.38 |

In order to mitigate the impact of file loading on the runtime, we performed 100 iterations on a same image of the Urban100 dataset (256×161 for the low resolution image as input) so as to eliminate any potential discrepancies. As to the GPU_InplaceGelu version, we replace the gelu action function with the InplaceGelu [3] to further improve the speed of model.

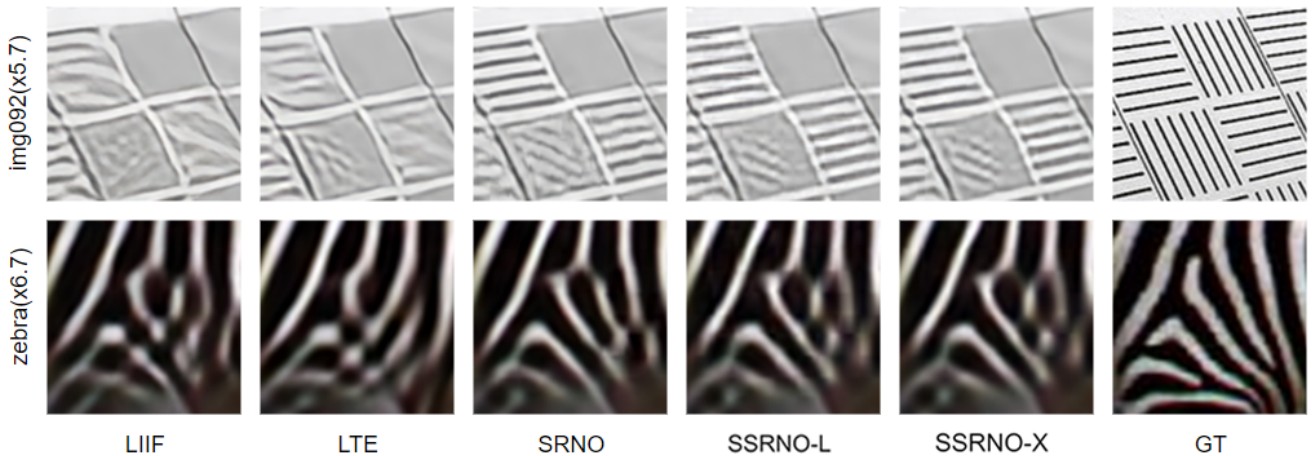

Figure 5: Visual comparison on other pictures on high-frequency. We choose the img092 from Ubran100[21] and zebra from Set14[44].

8.5 times FLOPS compared with the largest model SSRNO-X (see Table 4 for details).

We also test the through-put of our model shown in Table 4. Our model has the minimum FLOPs while running 2× faster than SRNO on both Nvidia RTX 4090 GPU and Intel Xeon CPU. We also convert the models to the ONNX format. Our model also runs nearly 2× faster than SRNO[42] in the ONNX versions. Employing the inplace-gelu method[3] to overcome the performance limitations of GELU[18], our model runs 2.3× faster than SRNO. According to table 4, our model is more sensitive to the performance change of GELU which indicates that our performance bottleneck primarily lies in GELU rather than the attention computations.

We further test our model on the evaluation datasets of Urban100, B100, Set5 and Set14. As shown in table 3, SSRNO performs better than EDSR-baseline-LTE[24] and closer to SRNO[42] on both in-distribution and out-of-distribution tests.

## 4.3 Qualitative result

Fig 4 illustrates the visualization results of two images from Urban100 dataset[21]. Both of the pictures are rich in high-frequency contents. All models perform well in recovering low-frequency information such as sunlight and buildings. However, SRNO and SSRNO demonstrate superior performance when it comes to higher-frequency features such as railings and texts. One can clearly observe that LIIF and LTE exhibit distortions in the railings, while SSRNO and SRNO are close to the ground truth in better preserving the railing's integrity.

Fig 5 shows two reconstruction comparisons on the high-frequency images patches from Urban100 and Set14 respectively. LIIF[10] and LTE[24] yield noticeable Moiré patterns on the zebra stripes, while SRNO and SSRNO produce clearer and less distorted textures. These results demonstrate that SSRNO still maintain the appealing performance of SRNO, even undergone significant compressions across different ratios.

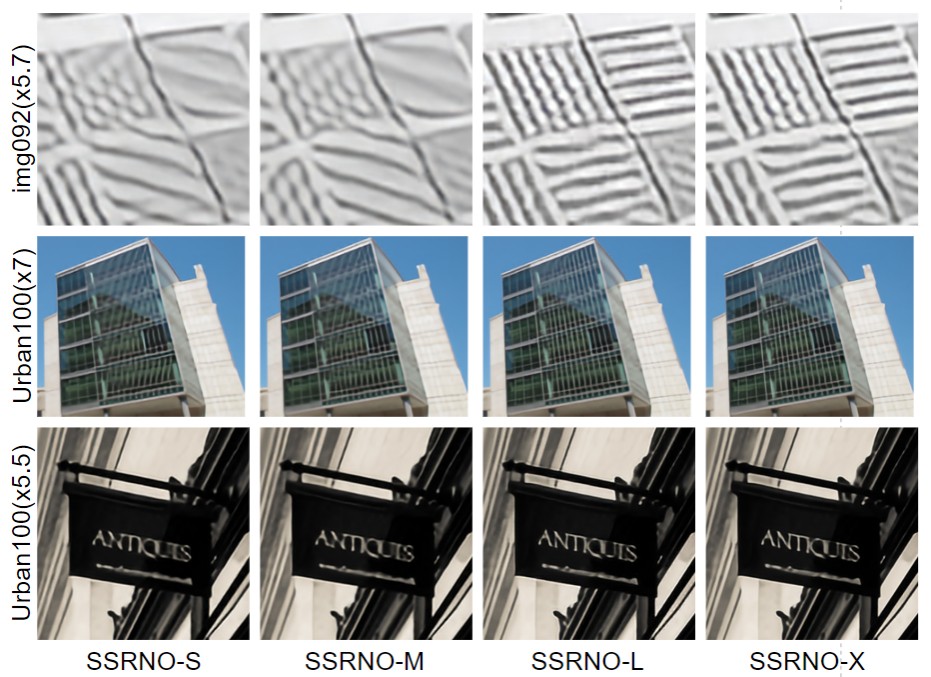

**Figure 6: Visual comparison on different model sizes.**

**Table 5: Encoder compression comparisons (PSNR (dB)).**

| Method | In-distribution | | Out-of-distribution |
|---|---|---|---|
| | ×2 | ×4 | ×6 |
| | Set5 | | |
| SRNO-INN-Encoder | 38.00 | 32.25 | 28.83 |
| SSRNO-Encoder | 38.03 | 32.24 | 28.87 |
| | Set14 | | |
| SRNO-INN-Encoder | 33.67 | 28.64 | 26.46 |
| SSRNO-Encoder | 33.69 | 28.64 | 26.48 |
| | B100 | | |
| SRNO-INN-Encoder | 32.18 | 27.59 | 25.84 |
| SSRNO-Encoder | 32.19 | 27.60 | 25.85 |
| | Urban100 | | |
| SRNO-INN-Encoder | 32.17 | 26.19 | 23.83 |
| SSRNO-Encoder | 32.25 | 26.23 | 23.87 |

SSRNO-Encoder refers to change the encoder part with ACRE method. We adjust SSRNO-Encoder with the same model size as SSRNO-INN-Encoder and compare their performance.

In order to verify the merit of model scalability, we show in Fig.6 comparisons of results from four compressed models, where one can observe the noticeable improvement in visual quality as the model size increases. This experiment clearly demonstrates the benefits SSRNO can bring when deployed on devices by virtue of its scalability and adaptability.

## 4.4 Ablation Study

As shown in Table 1 and Table 3 in SRNO-INN-Attention version, we remove the ACRE method and the modification of Galerkin-type attention to examine the effectiveness of our method. Compared with the SRNO-INN-Attention version, SSRNO-X performs better and achieves a better compression on the attention part. Furthermore, SRNO-INN-Attention version is slower than the SSRNO-X according to Table 4. These improvements show that our method is effective and better than simply using INN.

As the original INN can only be applied in convolutional and fully connected layers, for the sake of fair comparison, we test the ACRE method on the encoder part to provide a further verification of the effectiveness of it. Specifically, we use a compression rate as is recommended in the original INN paper. As results shown in Table 5, SSRNO-Encoder model using ACRE performs better than SRNO-INN-Encoder on all datasets and all output scales. This can demonstrate that, under the same model size, ACRE can allocate the parameters more reasonably, rather than blindly assigning the same amount of parameters to each layer.

## 5 CONCLUSION

In this paper, we propose an INN[39] representation for the Galerkin-type attention and an Adaptive Compression Rate Estimation (ACRE) algorithm which can help us to allocate model parameters more reasonably and determine the appropriate compression rates for each layer. With these two techniques we successfully demonstrate the continuous parameterization of SRNO, which allows for model scalability at inference time to adapt to different tasks and device resource constraints.

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
