# OpenReview forum: "Scalable Super-Resolution Neural Operator"
_acmmm.org/ACMMM/2024/Conference — MM2024 Poster_

### Official Review · Reviewer_3TKE · 2024-05-15

**Rating:** 2
**Confidence:** 3

**Summary:**

This work proposes a novel scalable super-resolution method named SSRNO that has faster
inference speed compared with existed methods. This method applies the INN to Galerkin type
attention to accelerate this module. Additionally, it proposes an adaptive layer-wise
compression rate estimation mechanism that sets the compression rates for different layers
adaptively.

**Strengths:**

1. The writing is clear and easy to understand.
2. This paper provides a series of well-designed ablation experiments that effectively illustrate
the role of each component within the proposed method.

**Limitations:**

1. This work employs the singular value to represent the importance of parameter matries. The
authors should provide more experiments results to validate the efficacy of this approach.
For example, many studies on neural network pruning utilize L1 or L2 norm as the
importance score of layers. It remains to be seen whether using singular values as
importance scores yields superior results compared to traditional L1 or L2 norm.
2. The novelty of the Scaler Galerkin-type Attention is questionable. The primary elements of
Galerkin-type Attention are linear layers, and the original INN framework has already
introduced a continuous parameter representation for these layers. Consequently, extending
this representation to Galerkin-type Attention may seem straightforward.

**Suitability:**

2

---

### Official Review · Reviewer_BkFc · 2024-05-24

**Rating:** 4
**Confidence:** 2

**Summary:**

This paper introduces a method called Scalable Super-Resolution Neural Operator (SSRNO)  for arbitrary scale SR modules. It extends the parameterization of SRNO with two innovative contributions: first, the Integral Neural Network (INN) formulation for Galerkin type attention, crucial for maintaining spatial discretization invariance in SR neural networks; second, an adaptive layer-wise compression rate estimation mechanism that adjusts to varying capacities across neural network layers. Extensive experiments validate the outperforming overall performances over existing continuous SR models.

**Strengths:**

1. This paper is well motivated as it addresses the deployment of deep learning models on platforms with dynamically constrained resources.
2. This research focuses on compressing the attention mechanism, a challenging yet valuable issue that contributes to the advancement of Transformer models.

**Limitations:**

1. The writing of this paper is poor due to its illogical structure, with large gaps within paragraphs (e.g., lines 859-863), undefined symbols (e.g., mats in Algorithm 1), and numerous grammatical issues.
2. Figures 2 and 3 in this paper are too simplistic, failing to effectively convey the authors' intentions. Additionally, many aspects are unclear (e.g., the meaning of the circle between the two plus signs in Figure 2).
3.In the experiments, this work's advantage over the previous work, SRNO, is very limited, making it difficult to highlight their contribution.

**Suitability:**

3

---

### Official Review · Reviewer_mRzS · 2024-05-27

**Rating:** 3
**Confidence:** 1

**Summary:**

This paper presents SSRNO, a method that adjusts the size of networks for improving image quality in super-resolution tasks. It builds on an existing method called SRNO[42], but adds two important updates: (1) a new way to focus on important features using what’s called Integral Neural Network (INN) for Galerkin-type attention, and (2) a smart system to estimate how much to compress each layer of the network, making it work better on devices with limited resources. Extensive testing shows that SSRNO not only works faster and fits better on small devices but also keeps the image quality high. It can make the model up to 62% smaller and more than twice as fast with barely any loss in image quality.

**Strengths:**

1. SSRNO introduces new methods like Integral Neural Network (INN) for Galerkin-type attention and a layer-wise compression rate estimation, enhancing the focus on relevant features and optimizing network size. As shown in Figure 1, SSRNO is better at achieving higher SR throughput with relative smaller sized network models comparing to most existing baselines.

2. The adaptive network size adjustment allows for efficient deployment on devices with limited resources. This approach offers a versatile solution where a model is trained just once but can provide various super-resolution configurations during inference and deployment.

**Limitations:**

1. The foundational approach of adaptive network size for super-resolution was initially established by SRNO[42]. This paper makes modest enhancements by (1) employing the Integral Neural Network (INN) to more effectively implement Galerkin-type attention, and (2) introducing a layer-specific compression estimator. Rather than offering a novel solution, this work seeks to optimize the efficiency of the existing model.

2. The explanations of some figures could be improved for better clarity. For example, Figure 2 does not clearly explain the training and inference workflow in SSRNO. Additionally, the concepts related to Neural Operators are not well-defined. I need refer to the SRNO[42] paper for a clearer understanding.

**Suitability:**

3

---

### Official Review · Reviewer_ndsH · 2024-05-27

**Rating:** 4
**Confidence:** 2

**Summary:**

This paper proposes an integral neural network representation for the Galerkin-type attention and an adaptive compression rate estimation algorithm to allocate model parameters more reasonably and determine the appropriate compression rates for each layer. It allows for model scalability at inference time to adapt to different tasks and device resource constraints. Experiments on multiple datasets demonstrate the effectiveness of the proposed method.

**Strengths:**

1. The inference-time adaptive network is meaningful for deploying SR models in resource-constrained devices.
2. Both qualitative and quantitative results are provided to illustrate the advantages of the proposed method.

**Limitations:**

1. The comparison with efficient SR models should be added in Fig.1.
2. How about the subjective evaluation, e.g., NIQE, LPIPS?
3. How to set the thresholds? More analysis should be provided.
4. In Table 4, are there the results of total runtime or average runtime in Urban100?
5. There are some writing typos which should be carefully corrected, e.g.,
- In Line 86, methoda -> method
- In Line 346, Eq.9 -> Eq.(9)
- In Line 442, FFN layers..

**Suitability:**

2

---

### Meta-Review · Area_Chair_QPGd · 2024-07-06

**Recommendation:** Accept (Poster)
**Confidence:** 4

**Metareview:**

In this paper the authors propose the Scalable Super-Resolution Neural Operator (SSRNO) to adjust the size of the networks for super resolution tasks with the INN formulation of Galerkin type attention and an adaptive compression rate estimation for model scalability.

The reviewers agree that there are several strengths in the paper (well motivated, well-written, inference-time adaptive network,  qualitative and quantitative results, novelty through INN, and athe adaptive network size adjustment, etc). The reviewers have also identified several weaknesses (more experiments needed, modest enhancements to baseline - some novelty issues, some editorial/structural issues).

Based on three positive reviews, it is recommended that the paper be accepted as a poster assuming that the suggested changes (esp. editorial and structural issues) be completed before the camera-ready deadline.